# Effects of two different emotion-inducing methods on the emotional memory of non-clinically depressed individuals

Wuji Lin[1,2,3,4], Jingyuan Lin[5], Xiaoqing Cai[1,2,3,4], Jun Deng[1,2,3,4], Yuan Gao[1,2,3,4], Lei Mo[1,2,3,4]*

**1** Key Laboratory of Brain, Cognition and Education Sciences, South China Normal University, Ministry of Education, Guangzhou, Guangdong, China, **2** School of Psychology, South China Normal University, Guangzhou, Guangdong, China, **3** School of Psychology, Center for Studies of Psychological Application, Guangzhou, Guangdong, China, **4** School of Psychology, Guangdong Key Laboratory of Mental Health and Cognitive Science, Guangzhou, Guangdong, China, **5** School of Psychology, Shenzhen University, Shenzhen, Guangdong, China

☯ These authors contributed equally to this work.
* molei@m.scnu.edu.cn

**Data Availability Statement:** All relevant data are within the paper and its Supporting information files.

## Abstract

In the study of emotional memory bias in depressed individuals, most previous studies have used emotional materials, but there were significant differences in the effects of different emotion-inducing methods on face memory. In the present study, two experiments were conducted to explore the effects of different emotion-inducing methods on memory between healthy participants and non-clinically depressed participants. The results from experiment 1 showed that when feedback was used as induction, the memory performance of the non-clinical depression group was significantly higher than that of the healthy group under the condition of negative feedback. Under positive and neutral feedback, there were no significant differences between the two groups. In experiment 2, when emotional materials were used as a mode of induction, no significantly difference in each emotional condition between the healthy and depressed groups was found. The results of the present study show that different methods of emotional induction have different effects on depressed participants. Compared with the emotion induced by the emotional material, the non-clinical depressed participants had a better memory effect induced by negative emotional events.

## Introduction

Numerous studies have shown that recall or recognition of depression-related materials in depressed individuals has a relatively obvious bias [1–3], which is called the negative mood-congruent effect [4–6]. These cognitive biases in depressed individuals may lead to the onset or deepening of depression [3, 7]. According to Beck's schema theory, negative bias of memory and attention in depressed individuals can be explained by the fact that depressed individuals have a depression schema [8]. When this latent depression schema is activated by stressful events, it will lead to negative cognition, which in turn, leads to depression [6, 9, 10].

**Funding:** This study was supported by grants from National Social Science Fund of China [19ZDA360] and China Postdoctoral Science Foundation [2020M672660].

**Competing interests:** The authors have declared that no competing interests exist.

Most previous studies used emotional materials as induction methods to study the emotional memory of depressed individuals. For example, Ridout et al. (2003) required depressed patients and non-depressed patients to complete a facial emotion identification task and then a delayed recognition memory task [11]. The two groups categorized the emotional faces similarly. However, the groups had different memories concerning different types of faces. The depressed group recognized more negative faces and fewer positive faces than neutral faces. The non-depressed group had a better memory of positive faces and a worse memory of negative faces. Haque (2014) used the autobiographical memory test (AMT) to study the memory retrieval sequence of depressed patients. The results showed that depressed participants recovered autobiographical memory faster, produced shorter memory descriptions and were less likely to recall positive memories than non-depressed participants [12]. There are also many studies on directed forgetting that use emotional materials to induce emotions [13, 14]. These studies have consistently found that depressed participants recall fewer positive items than non-depressed participants, but they can recall more negative and neutral items. However, some studies did not find negative bias in the memory of emotional materials in depressed individuals [5, 15, 16]. For example, Ridout and Dritschel (2009) studied whether a memory bias of a sad face occurred in severely depressed participants after a random coding task. In the coding phase, depressed and non-depressed participants were asked to observe faces with different valences and judge each face's gender. Then, participants completed a recognition memory test. The results showed that the depression group did not show a negative mood-congruent memory bias in the recognition of sad faces [15].

Most studies use emotional materials as induction methods to study the emotional memory bias of depressed individuals. However, there are significant differences in the effects of different emotion-inducing methods on memory. In research using emotional materials, an obvious emotionally enhanced memory (EEM) has been observed; that is, emotional information can be remembered more easily than neutral information [17–19] because negative stimuli narrow attention and allow participants to remember details well, thus enhancing memory. On the other hand, positive stimulation broadens attention, thus enhancing gist memory [20–23]. For example, in a study by Kensinger et al. (2008), positive, negative and neutral pictures were used as experimental materials to explore the effects of valences of emotion on memory; they subsequently demonstrated that both positive and negative emotions showed stable EEM [23].

In contrast, significantly different results were observed in a study of emotional events as emotion-inducing methods [24–27]. For example, in a study by Mather et al. (2011), during the coding phase, participants were told that the response to the current stimulus would lead to gain, loss, or a draw [24]. The participants were subsequently asked to judge whether a picture was positive or negative and finally presented with feedback. In the test phase, participants were asked to recall and recognize the pictures. In both the recall and recognition tasks, the memory effect of the gain condition was better than that of the loss condition. Researchers believe that this finding was due to midbrain dopamine regions responding to motivationally relevant information interacting with the hippocampus to enhance memory for that information [28, 29]. Lin et al. (2020) compared the effects of feedback-induced emotion and material-induced emotion on memory. The results showed that in the case of emotion caused by feedback, positive feedback enhanced the memory of faces, while negative feedback decreased memory performance. In the case of emotion caused by emotional materials, the memory performance under negative and positive conditions was significantly higher than that under neutral conditions, and the memory performance under negative conditions was higher than that under positive conditions [27].

Different emotion-inducing methods have demonstrated different effects on memory. Therefore, we desired to investigate whether there is a difference between depressed and

healthy individuals. Is there any difference between these two emotion-inducing methods of negative mood-congruent bias? In the present study, two experiments were conducted to explore whether distinct negative mood-congruent effect produced differences in face memory between non-clinically depressed individuals and healthy individuals. We used the paradigm described in Lin et al. (2020) [27]. In the first experiment, the participants' guess of neutral faces were used as the conditions for inducing emotional events. Different emotional conditions were formed by connecting the feedback results with neutral faces to explore whether there was a difference in the effect of emotional events on memory between depressed individuals and healthy individuals. In the second experiment, faces expressing different emotions were used as the emotion-inducing material to explore whether there were differences in the effects of the emotional materials on memory in the two groups.

## Experiment 1: Effects of feedback results on memory

### Methods

**Ethics statement.** The procedure in this study was approved by the ethical review board of the School of Psychology, South China Normal University.

**Participants.** We recruited participants at a university in China. All participants were required to complete the Chinese version of the Beck Depression Inventory II (BDI-II-C) before being invited to participate. Previous research has shown that when using the BDI-II-C to screen Chinese individuals for depression, 14 points is the most effective threshold [30]. Therefore, in the present study, participants with BDI-II-C scores greater than or equal to 14 were selected as the depression (non-clinical) group, and participants with scores less than or equal to 4 were selected as the healthy group. There were 30 participants in the depression group and 30 participants in the healthy group, providing a total of 60 participants (35 women), aged between 18 and 25 years. All participants reported that they were not diagnosed with clinical depression or using anti-depressants. Additionally, they had no other comorbidities or used other psychotropic drugs. This study was approved by the local ethics committee, and all participants gave written informed consent in each experiment.

**Stimuli.** A total of 160 neutral faces, of which 80 were men and 80 were women, were selected from the Chinese Facial Affective Picture System [31]. This system also provided the valences and arousal data for all of the selected face images. The valences and arousal were rated by 60 participants in the study by Wang et al. (2005) [31]. The pixel size of the face images was $260 \times 300$. To investigate whether the faces correlated to the positive, negative, and neutral feedback manipulations during the learning phase, we randomly allocated 160 neutral faces into four groups: three of which consisted of the old faces related to positive, negative, or neutral feedback conditions during the learning phase, and the remaining group consisted of new faces mixed with old faces and was displayed during the memory test phase. There were 40 faces in each group; half were men, and half were women. In the learning phase, the faces were randomly presented in pairs (the same gender in each pair), and the two faces were aligned horizontally and fixed 20 cm apart. In the test phase, three groups of old faces and one group of new faces were randomly and individually presented. The ratio of old-to-new faces was 3:1. Analyses of valence and arousal ratings show that the 4 groups of faces did not systematically differ from each other. One-way ANOVA (condition: positive, negative, neutral, new) was performed on the valences. The main effect of condition was not significant [$F(3,156) = 0.789$, $p = 0.502$; positive: $4.313 \pm 0.586$; negative: $4.174 \pm 0.524$; neutral: $4.222 \pm 0.472$; new: $4.320 \pm 0.425$]. One-way ANOVA of arousal (condition: positive, negative, neutral, or new) showed that the main effect of face type was not significant [$F(3,156) = 1.252$, $p = 0.293$; positive: $3.609 \pm 0.512$; negative: $3.816 \pm 0.632$; neutral: $3.841 \pm 0.717$; new: $3.875 \pm 0.815$].

**Procedures.** The experimental program was compiled by Presentation 0.71 software. The participants sat on seats in a soundproof room to complete the experiment. The background of the monitor was black, and the distance of the screen from the participants was 80 cm.

Learning phase: There were two faces presented simultaneously, one on the left and one on the right side of the screen. The presentation time was800 ms, and the participants were required to judge the identity of the faces. To avoid the cheater effect [32], half of the participants were asked to judge which face was the cheater and the other half were asked to judge which face was trustworthy. If participants selected the left face, they pressed the F key. If participants selected the right face, they pressed the J key. The feedback results were given after the keyboard was pressed. Then, the next pair of faces was presented at an interval of 1400-1800ms. All participants were told that, "There are two situations: in one, one face is a cheater and the other is trustworthy; in the other, the two faces are trustworthy. In the first case, when the judgment is correct, "correct" is presented in the feedback stage; when the judgment is in error, "wrong" appears in the feedback stage. In the second case, there is a "draw" in the feedback stage. The participants initially had 100 points; when the judgment was correct, 1 point was added, and when the judgment was incorrect, 1 point was subtracted; in a draw, the score remained unchanged. The reward was calculated based on the final score." If participants feel both faces were trustworthy (or cheater), they must make a force-choice. To ensure that there were a sufficient number of faces in each group, each pair of faces was in a fixed feedback group; that is, in the positive group, regardless of which face the participants chose, the feedback was "correct", and in the negative group, the feedback was "incorrect". The feedback for the neutral group was "draw". The two simultaneously presented training faces were tagged as negative following a "wrong" feedback (or as positive following a "correct" feedback or as neutral following a "draw" feedback) and both were used in the recognition test.

Test phase: A single face was presented in the center of the screen, and the participants were asked to judge whether the presented face was previously unseen. If participants judged it as new, they pressed the F key. If participants judged it as old, they pressed the J key. The button response was counterbalanced across participants. The face remained displayed until the participant responded, and then the next face was presented at intervals of 1400–1800 ms.

**Analysis.** The analysis was performed using SPSS 24.0. A measure of old/new discrimination (Pr) was computed: $Pr = p(\text{hit}) - p(\text{false alarm})$ based on the work by Snodgrass and Corwin (1988) [33]. Hit represents when a participant indicated an old (learned) face as an old one. False alarm represents when a participant indicated a new face as an old one. The reaction time represented the average response time in trials with correct responses.

A mixed-design ANOVA was performed with a within-participants factor with 3 levels (feedback type: positive, negative, or neutral) and a between-participants factor with two levels (group: depression, healthy). The least significant difference (LSD) was used for post hoc comparisons.

## Results

The results of Pr values showed that the main effect of feedback type was significant [$F(2, 116) = 6.776$, $p < 0.01$, $\eta p^2 = 0.105$]. A post hoc test revealed that the recognition Pr value of positive feedback was significantly higher than that of negative feedback [$p < 0.01$], and neutral feedback [$p < 0.01$]. No significant difference was found concerning the recognition performance between negative and neutral feedback conditions [$p = 0.822$]; the main effect of group was not significant [$F(1, 58) = 0.689$, $p = 0.410$, $\eta p^2 = 0.012$]. Importantly, the interaction of feedback type and group was significant [$F(2, 116) = 3.578$, $p < 0.05$, $\eta p^2 = 0.058$]. Furthermore, simple effect analysis showed that the performance in the healthy group differed from the

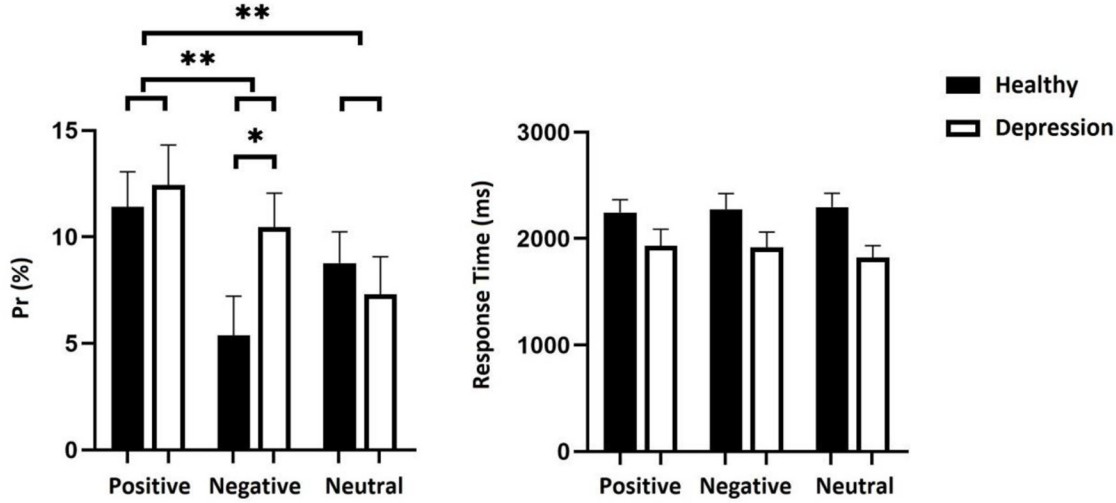

**Fig 1. Pr and mean RT in Experiment 1.** Error bars represent standard errors. $^{*}p<0.05$; $^{**}p<0.01$; $^{***}p<0.001$.

depressed group in the negative feedback condition [$t(58) = -2.119$, $p< 0.05$], and the mean value of Pr in the depressed group was higher than that in the healthy group. The difference between healthy and depressed was not significant in positive feedback [$t(58) = -0.441$, $p = 0.661$] and neutral feedback conditions [$t(58) = 0.648$, $p = 0.520$].

The results of RTs showed that the main effect of feedback type was not significant [$F(2, 116) = 0.200$, $p = 0.819$, $\eta p^2 = 0.003$], and the interaction between feedback type and group was not significant [$F(2, 116) = 0.897$, $p = 0.411$, $\eta p^2 = 0.015$]. Only the main effect of the group was significant [$F(1, 58) = 4.714$, $p< 0.05$, $\eta p^2 = 0.075$], and the reaction time of the healthy group was significantly longer than that of the depressed participants. (See Fig 1).

## Discussion

Experiment 1 explored the effects and differences of feedback as a method of emotional induction in depressed and non-depressed participants. First, the Pr value was highest under the positive condition and lowest under the negative condition. The results of previous studies also show that the memory effect of memory items was greatest after receiving positive feedback, similar to a reward, while the memory effect was the worst after receiving negative feedback, similar to punishment [24, 34]. Consistent with previous studies [24, 27], positive feedback enhances memory, while negative feedback weakens memory. Lin et al. believe that this phenomenon occurs because positive feedback enhances memory consolidation and is conducive to individual behavior that is rewarded repeatedly. On the other hand, negative feedback inhibits the formation of memory to avoid the harm of negative emotion to the individual.

In the comparison of non-clinically depressed participants and healthy participants, there was no significant difference between the positive condition and neutral condition. On the other hand, there was a significant difference under the negative condition; that is, the memory effect of the depressed group was greater than that of the healthy group under negative feedback. Similar results have been found in previous studies. In the study of Haque et al. (2014), the researchers gave five categories of words as cues, including common locations, general objects, positive emotions, negative emotions and important other words. The participants were asked to recall the events associated with the cues. The results showed that, compared

with the non-depression group, the depression group recalled fewer positive events and more negative and neutral events [12]. In the study of Rottenberg et al. (2006), the researchers asked participants to describe the happiest and saddest events in detail and found that it was easier, more specific and more emotional for the depression group to recall sad events than happy events [35]. Thus, depressed individuals have demonstrably better memory of negative events, which may be explained by the concept that negative events are consistent with the depression schema. The discovery of negative events activates the depression schema, which, in turn, promotes the memory of negative events.

## Experiment 2: Effects of emotional expressions on memory

### Methods

**Ethics statement.** The procedure in this study was approved by the ethical review board of the School of Psychology, South China Normal University.

**Participants.** There were 30 participants in the depression group and 30 participants in the healthy group, with a total of 60 participants (36 women), aged between 18 and 26 years old. The screening and grouping criteria of the participants in the depression group and the healthy group were the same as those in experiment 1. The participants who participated in experiment 1 did not participate in experiment 2. This study was approved by the local ethics committee, and all participants gave written informed consent in each experiment.

**Stimuli.** Eighty negative faces, 80 neutral faces, and 80 positive faces from the Chinese Facial Affective Picture System were selected; half were men and half were women. The emotional faces were randomly assigned to old (60 faces) and new (20 faces) conditions, of which half were men and half were women. The experiment was divided into two phases as follows: the learning phase and the testing phase. In the learning phase, the faces were randomly presented in pairs (the same gender and same emotional expression in each pair). In the test phase, three groups of old faces and one group of new faces were randomly and individually presented. The ratio of old-to-new faces was 3:1.

The valences and arousal data was provided by Chinese Facial Affective Picture System. A two-way ANOVA with 3 (valence: positive, negative, or neutral)×2 (face type: new and old) factors revealed that the main effect of valence was significant [$F(2,234) = 291.425$, $p<0.001$], the main effect of face type was not significant [$F(1,234) = 0.581$, $p = 0.447$], and the interaction effect was not significant [$F(2,234) = 0.850$, $p = 0.429$, negative old: $2.855 \pm 0.469$; negative new: $3.042 \pm 0.664$; neutral old: $4.267 \pm 0.557$; neutral new: $4.171 \pm 0.563$; positive old: $5.629 \pm 0.814$; positive new: $5.752 \pm 0.561$]. A two-way ANOVA with 3 (valence: positive, negative, or neutral) × 2 (face type: new and old) factors for arousal was performed and revealed that the main effect of valence was significant [$F(2,234) = 86.915$, $p<0.001$], the main effect of face type was not significant [$F(1,234) = 0.427$, $p = 0.514$], and the interaction effect was not significant [$F(2,234) = 0.166$, $p = 0.847$] (negative old: $6.089 \pm 1.256$; negative new: $6.300 \pm 1.059$; neutral old: $3.689 \pm 0.532$; neutral new: $3.783 \pm 0.720$; positive old: $4.705 \pm 1.176$; positive new: $4.701 \pm 1.154$).

**Procedures.** The experimental procedure was programmed using Presentation 0.71 software. The participants sat in a seat in a soundproof room to complete the experiment. The background of the monitor was black, and the monitor was placed 80 cm away from the participant.

*Learning phase.* The procedures of this experiment were identical to that of experiment 1, with one exception; no feedback was provided after the button was pressed. To be specific, there were two faces, one on the left and one on the right side of the screen; the presentation time was 800 ms, and the participants were required to judge the identity of the faces. If

participants selected the left face, they pressed the F key. If participants selected the right face, they pressed the J key. The feedback results were not given after the keyboard was pressed. Then, the next pair of faces was presented at an interval of 1400-1800ms. All participants were told that "There are two situations: in one, the one face is a cheater, and the other is trustworthy; in the other, the two faces are trustworthy. The participants initially had 100 points; in the first case, when the judgment was correct, 1 point was added, and when the judgment was incorrect, 1 point was subtracted; in the second case, the score remained unchanged. The reward was calculated based on the final score."

*Test phase*. A single face was presented in the center of the screen, and the participant was instructed to judge whether the face had appeared previously during the learning phase. If participants judged it as new, they pressed the F key. If participants judged it as old, they pressed the J key. The button response was counterbalanced across participants. The face remained displayed until the participant responded, and then the next face was presented at intervals of 1400–1800 ms.

**Analysis.**   All faces in experiment 1 were neutral faces. Different conditions are divided by feedback results. New faces in the test phase were neutral and did not need to be divided into three conditions. Therefore, there was no difference in the false alarm rate among the conditions. In contrast to experiment 1, different conditions are divided by expression in experiment 2. If all of the new faces were neutral, it would be too easy for participants to judge the positive or negative faces as old. Therefore, new faces of the three conditions represented three kinds of expression, and the false alarm rates in the three conditions may have been different. Therefore, in experiment 2, the Pr value, reaction time, hit rate, and false alarm rate were analyzed.

Mixed-design ANOVAs of the new and old recognized Pr values, hit rate, false alarm rate and RTs were performed with 3 (valence: positive, negative, or neutral) ×2 (group: depression and healthy) factors. The LSD was used for post hoc comparisons.

## Results

The results of the Pr values showed that the main effect of material valence was highly significant [$F(2, 116) = 101.659, p < 0.001, \eta p^2 = 0.637$]. A post hoc test revealed that the negative was significantly greater than the positive [$p < 0.001$], the positive was greater than the neutral [$p < 0.001$], and the negative was significantly greater than the neutral [$p < 0.001$]. The main effect of the group was not significant [$F(1, 58) = 0.061, p = 0.806, \eta p^2 = 0.001$]. The interaction between valence and group was not significant [$F(2, 116) = 0.338, p = 0.714, \eta p^2 = 0.006$].

In terms of the hit rate, the main effect of valence was significant [$F(2, 116) = 17.051, p < 0.001, \eta p^2 = 0.227$]. A post hoc test revealed that the negative was significantly greater than the positive [$p < 0.05$]; the negative was also significantly larger than the neutral [$p < 0.001$], and the positive was significantly greater than the neutral [$p < 0.001$]. The main effect of the group was not significant [$F(1, 58) = 0.028, p = 0.868, \eta p^2 = 0.000$]. The interaction between valence and group was not significant [$F(2, 116) = 0.997, p = 0.372, \eta p^2 = 0.017$].

In terms of the false alarm rate, the main effect of valence was significant [$F(2, 116) = 56.488, p < 0.001, \eta p^2 = 0.476$]. A post hoc test revealed that the negative was significantly less than the positive [$p < 0.001$], the negative was also significantly less than the neutral [$p < 0.001$], and there was no significant difference between positive and neutral [$p = 0.280$]. The main effect of the group was not significant [$F(1, 58) = 0.118, p = 0.732, \eta p^2 = 0.002$]. The interaction between valence and group was not significant [$F(2,116) = 0.239, p = 0.788, \eta p^2 = 0.004$].

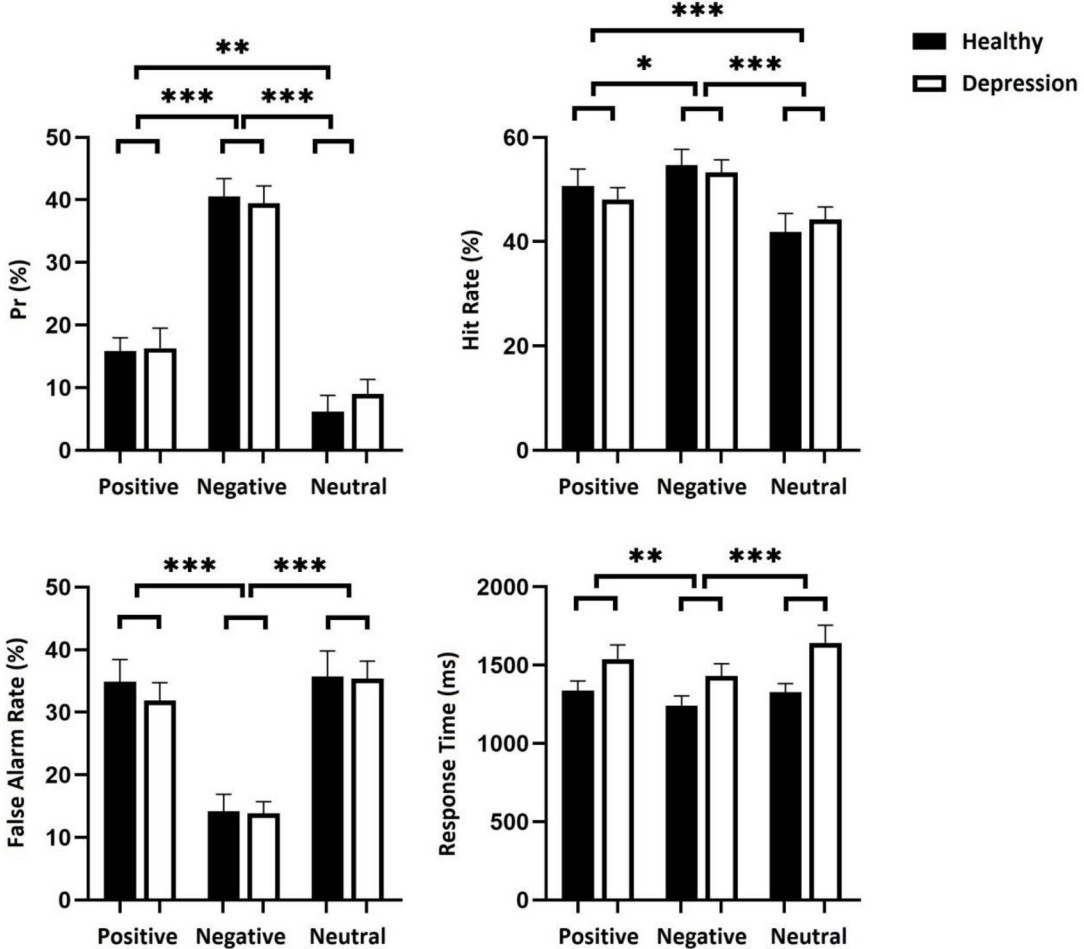

**Fig 2. Pr, hit rate, false alarm rate and mean RT in Experiment 2.** Error bars represent standard errors. $^*p<0.05$; $^{**}$ $p<0.01$; $^{***}$ $p<0.001$.

The results of RTs showed that the main effect of material valence was significant [$F(2,116)$ = 10.629, $p< 0.001$, $\eta p^2$ = 0.155]. A post hoc test revealed that the negative was significantly less than the positive [$p< 0.001$], the negative was also significantly less than the neutral [$p< 0.001$], and the difference between positive and neutral was not significant [$p = 0.180$]. The main effect of the group was significant [$F(1, 58)$ = 5.005, $p< 0.05$, $\eta p^2$ = 0.079]. A post hoc test revealed that the reaction time of the healthy group was significantly lower than that of the depression group [$p< 0.05$]. The interaction between valence and group was not significant [$F(2, 116)$ = 2.305, $p = 0.104$, $\eta p^2$ = 0.038]. (See Fig 2).

## Discussion

The aim of the second experiment was to explore the influence and difference in memory effects induced by emotional materials between depressed individuals and healthy individuals. First, the hit rate under the negative condition was greater than that under the positive condition, and the hit rate under the neutral condition was the lowest. Concerning false alarm rates,

there was no significant difference between the neutral and the positive conditions, and both were significantly greater than those of the negative condition. The results of reaction time and Pr value showed that the memory effect of negative faces was the best, that of positive faces was the second best, and that of neutral faces was the worst, which is consistent with the results of most previous studies, reflecting the EEM [13, 36–38].

In the comparison of the results of depressed participants and healthy participants, there was no significant difference in Pr value and hit rate between depressed participants and healthy participants; however, there was a significant difference in RTs, and the RTs of depressed participants were longer. When emotional and non-emotional information are processed in parallel, information-processing biases could occur as a function of attention to affective information [39, 40]. This is called affective interference. Compared with healthy individuals, depressed individuals react slowly to information because depressed individuals are sensitive to affective interference [39]. Therefore, in the test phase of experiment 2, when participants did the non-emotional task when presented with the emotional faces, the depressed participants allocated more cognitive resources to emotion. It caused the dispersion of cognitive resources of non-emotional tasks [3, 41, 42]. So, the reaction time of depressed participants was longer than that of healthy participants.

## General discussion

In the present study, two experiments were conducted to compare the effects of two emotion-inducing methods on face memory between depressed and healthy participants. The results show that the two emotion-inducing methods have different effects on memory performance. In the experiment using feedback as an induction method, the memory effect of faces connected with positive feedback was determined to be the best, while that of faces connected with negative feedback was the worst. In the negative feedback group, the memory performance of the depression group was better than that of the healthy group, but there was no significant difference between the neutral and positive feedback groups. In experiment 2, using emotional materials as a mode of induction, the memory effect of negative faces was the best, while that of neutral faces was the worst. Although there was no significant difference in Pr value or hit rate between depressed individuals and healthy individuals, the reaction time of the depression group was longer.

The two experiments show different experimental results. The first possible explanation is due to the different cognitive processing methods [27]. For the emotion induced by feedback, the emotional event of the feedback is tied to the memory item by the experimental task in the coding phase. In the retrieval phase, the retrieval of memory items will also activate the emotions tied to memory items. Therefore, under the condition of positive feedback, positive emotion will enhance the memory effect to repeat the rewarded behavior. Under the condition of negative feedback, the inhibition of negative events during retrieval can avoid the harm of negative emotion. The emotion induced by the emotional material is generated by the memory material itself, and the negative emotion cannot be weakened by inhibition in the retrieval phase. In contrast, negative stimuli narrowed attention and enhanced the participants' memories of details [20, 21]. Therefore, the memory performance from negative stimulation is better.

Different cognitive processing methods also lead to differences in the performance of the two emotion-inducing methods between depressed individuals and healthy individuals. In the present study, the negative mood-congruent effect appeared only in experiment 1 but not in experiment 2. Previous studies have found that patients with depression have a defect in inhibitory control in the phase of memory coding and retrieval [43, 44]. Joormann's theory of

cognitive control impairment assumes that because patients are unable to actively forget negative events, they will continue to retell and regurgitate negative emotional content, thus making it difficult to extricate themselves from a negative mood [45–47]. In experiment 1 of the present study, the participants with depression tended to ruminate on the memory items with negative feedback continuously to improve their memory effect. However, there was no continuous ruminating for the memory items with neutral and positive feedback. Therefore, under the negative condition, the memory performance of the depression group was higher than that of the healthy group. Under neutral and positive conditions, there was no significant difference between the groups. Many studies have found that individuals with depression have a stronger attention bias toward negative materials. It can further promote subsequent memory effects by enhancing the encoding and retrieval of sensory information [21, 22, 23, 48, 49]. However, in experiment 2 of the present study, no obvious negative emotional consistency effect was observed, which may have been because the experimental task of the present study was different from that of previous studies, and the guessing of faces reduced the processing of negative expressions.

Second, the mood types induced by the two modes were different [27]. The emotional experience caused by decision failure in experiment 1 is an emotion of regret, which occurs when we reject a better outcome and choose a worse outcome [50] and is regulated by a cognitive process called counterfactual thinking [51, 52]. The regret emotion in experiment 1 is different from the emotion that can be directly triggered by emotional stimulation in experiment 2, which is essentially a type of emotion based on complex cognition [53]. Previous studies have shown that even if they are also negative emotions, different emotional types will have different effects on memory [18, 54, 55]. Therefore, different emotional types may also lead to different memory performances.

Different emotional types may also lead to varying face memory performances between the two emotion-inducing methods comparing depressed and healthy individuals. The emotion of regret is regulated by counterfactual thinking, and the ruminating processing of negative events by depressed individuals may enhance this cognitive process, deepen the emotion of regret, and then enhance the negative mood-congruent effect of depressed individuals, as well as the effect of item memory associated with negative feedback. The emotion in experiment 2 was induced by emotional materials, and ruminating does not enhance the processing of emotional materials.

In summary, the negative mood-congruent effect does not exist consistently in all types of emotional memories. Furthermore, in individuals with a tendency towards depression, compared with the effect of negative emotional materials on memory, the emotion induced by negative feedback has a stronger effect, which may be due to the memory enhancement of depressed individuals through negative events. Therefore, weakening the memory of negative events in depressed individuals may be an effective means to reduce the degree of depression. In cognitive behavioral therapy (CBT), which is the most widely used treatment for depression, eliminating the negative prejudice of depressed individuals during information processing is the clear goal [56–58]. CBT generally takes a step-by-step approach, first teaching depressed individuals how to monitor their emotions, thoughts and behaviors. The next step is to promote their participation in activities that provide positive emotions, which is conducive to short-term mood regulation and provides an emotional environment for long-term treatment. The last step is to help depressed individuals test their automatic thinking and core cognitive patterns and teach them to reconstruct a correct cognitive structure [59]. Once the cognitive defects are changed and corrected, bad mood and behavior will improve accordingly [60–62].

Future studies can pay greater attention to the effects of different induction modes on the emotional memory of depressed individuals and use neuroscience techniques to understand

the brain mechanism, which will help understand some of the cognitive characteristics of depressed individuals and improve the related theory of depression.

## Supporting information

**S1 Data. The data of experiments.**
(ZIP)

## Author Contributions

**Conceptualization:** Wuji Lin, Jingyuan Lin.

**Data curation:** Wuji Lin.

**Formal analysis:** Wuji Lin.

**Funding acquisition:** Lei Mo.

**Methodology:** Wuji Lin, Jingyuan Lin, Xiaoqing Cai.

**Project administration:** Lei Mo.

**Supervision:** Lei Mo.

**Writing – original draft:** Xiaoqing Cai.

**Writing – review & editing:** Wuji Lin, Jingyuan Lin, Jun Deng, Yuan Gao.

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
