## [Decision Letter · Decision Letter 0]

3 Feb 2021

PONE-D-20-39095

Effects of two different emotion-inducing methods on the emotional memory of non-clinical depressed individuals

PLOS ONE

Dear Professor Mo,

Thank you for submitting your manuscript to PLOS ONE. After careful consideration, I feel that it has merit but does not fully meet PLOS ONE’s publication criteria as it currently stands. Therefore, I invite you to submit a revised version of the manuscript that addresses all points raised by both Referees.

We look forward to receiving your revised manuscript.

Kind regards,

Angela Sirigu

Academic Editor

PLOS ONE

2. Our internal editors have looked over your manuscript and determined that it is within the scope of our Cognitive Developmental Psychology Call for Papers. The Collection will encompass a diverse range of research articles in developmental psychology, including early cognitive development, language development, atypical development, cognitive processing across the lifespan, among others, with an emphasis on transparent and reproducible reporting practices.  Additional information can be found on our announcement page: https://collections.plos.org/s/cognitive-psychology.  If you would like your manuscript to be considered for this collection, please let us know in your cover letter and we will ensure that your paper is treated as if you were responding to this call. Please note that being considered for the Collection does not require an additional peer review beyond the journal’s standard process and will not delay the publication of your manuscript if it is accepted by PLOS ONE. If you would prefer to remove your manuscript from collection consideration, please specify this in the cover letter. 

3. Please ensure that you include a title page within your main document. We do appreciate that you have a title page document uploaded as a separate file, however, as per our author guidelines (http://journals.plos.org/plosone/s/submission-guidelines#loc-title-page) we do require this to be part of the manuscript file itself and not uploaded separately.

Reviewers' comments:

Reviewer's Responses to Questions

**Comments to the Author**

1. Is the manuscript technically sound, and do the data support the conclusions?

Reviewer #1: Yes

Reviewer #2: Yes

2. Has the statistical analysis been performed appropriately and rigorously? 

Reviewer #1: Yes

Reviewer #2: Yes

3. Have the authors made all data underlying the findings in their manuscript fully available?

Reviewer #1: Yes

Reviewer #2: Yes

4. Is the manuscript presented in an intelligible fashion and written in standard English?

Reviewer #1: Yes

Reviewer #2: Yes

5. Review Comments to the Author

Reviewer #1: Lin et al. investigated the impact of emotional induction on memory for faces in normal depressed participants, addressing the question whether a depressed profile is accompanied by heightened sensitivity for negative memories. Two facial recognition experiments were conducted, which differed according to the emotion-induction method employed. Experiment 1 used faces with neutral expressions that, during a training phase, were associated with negative, positive or neutral feedback and then tested recognition memory for these faces. Experiment 2 exposed participants to faces with negative, positive or neutral expressions during the training phase and evaluated whether recognition memory was affected by the intrinsic emotional quality of the faces. Very different results were obtained in the two experiments. In the first, normal participants show better memory for positive as compared to negative faces but depressed participants deviated from this pattern by not showing this “suppression” of memory for negative faces. In contrast, in Experiment 2 recognition memory was better for previously seen faces with negative as compared to positive and neutral expressions, and there were no differences between depressed and non-depressed participants, suggesting that memory performance on this task variant is enhanced by the emotional saliency/arousal value of the face stimuli and this, to the same extent in the two groups.

This study has merits from both the methodological and theoretical point of views and could make a useful contribution to the field. The testing procedures and statistics applied appear appropriate. However, several points need to be addressed/clarified and the text requires some improvements.

My comments are listed below.

1) What is the relevance of non-clinical depressed subjects for study aimed at understand cognitive/emotional processing in depression? Is there any evidence ofa continuum between non-clinical and clinical depression? That high BDI scores in non-clinical populations are a predictor of risk for depression? Since, in several parts of the paper, participants as referred to as “healthy” and “depressed”, it is important point to address this point, at least as a cautionary note in the interpretation of the data.

2) p.9 2nd par (“Stimuli”), line 2 : “All facial pictures in the system were obtained by all 60 participants” Unclear, please rephrase.

3) p.9 2nd par (“Stimuli”) : Who performed the valence rating of the pictures? The same 60 subjects that participated in the memory experiment or a separate group? If they are the same, when did they perform the rating?

4) p.10 1st par, line 2: Judging the “identity” is not exactly what participants were asked to do. It seems more like a personality or trustworthiness judgement was requested.

5) p.10 1st par, line 4: “The participants pressed a key to give feedback”. I think ‘to receive feedback’ is meant here, right? Also, as I understand it, on each face pair presentation, there were three possible options to choose from: left is a cheater, right is a cheater, none is a cheater. How did participants express their choice? Vocally? By pressing different keys?

6) p.10 1st par: As I understand it, although I’m not 100 percent sure, the two simultaneously presented training faces were tagged as negative following a “wrong” feedback (or as positive following a “correct” feedback) and both were used in the recognition test. Is that correct?

7) p.10 2nd par, line 2: “The face persisted until the reaction disappeared” Unclear. What do the authors means? That the face remained displayed until the participant responded? Please re-phrase more clearly.

8) p.10 4th par (Results): Several comparisons are made between healthy and “depression”. It should probably read “depressed”

9) p.12 4th par (Stimuli): The authors present the results of an analysis confirming the different valences of the test stimuli, but do not explain how the data were obtained.

10) p.13 2nd par: Here again, the reader is left to guess what the participants were instructed to do during the learning phase. Please be more explicit.

11) p.14 2nd par, 1st line: “The second experiment was”: I think the authors meant ‘The aim of the second experiment was’.

12) p.15 1st par, line6-7: I am skeptical about the interpretation of higher RTs in depressed participants: it is not obvious why, when the task is not to identify not the emotion, but to recall the familiarity of a face, a tendency to misinterpret emotions should slow down response times. Furthermore, if a bias to assign a negative emotion to neutral face did contribute to slow down RTs, why are these results opposite to those obtained in Experiment 1 with neutral faces only (i.e. faster RTs in depressed participants)? Actually, the opposite direction of the RT difference between normal and depressed participants between the two experiments is quite puzzling.

13) p.15 2nd par, line 3-4: “on the performance of memory”: ‘on memory performance’ wound read better.

14) p.16 1st par line 7: “individuals with depression have a stronger attention bias toward negative materials and the further promotion of subsequent memory effects” The second part of this sentence is unclear. Please rephrase.

Finally, the manuscript must be checked by an English language native speaker.

Reviewer #2: In the present study, memory for face items (recognition task) in non-clinically depressed participants and in controls was tested. In experiment 1, recognition performance was tested after participants received negative, positive or neutral feedbacks in the encoding phase. In experiment 2, the performance was evaluated for face items having positive, negative or neutral expressions. Results show that recognition in “depressed” participants was better for faces associated with a negative feedback, while no group differences were found in experiment 2. The study is well performed and the results are interesting. I have, however, a few comments that might help improve the manuscript.

Subjects were divided in two groups (healthy participants vs. non-clinical depression participants) based on BDI scores: “participants with BDI-II-C scores greater than or equal to 14 were selected as the depression group (non-clinical).” How do the authors know that the group of “depressed” participants did not include subjects with clinical/severe depression? Did the authors collected more information about the subjects’ medical history and were subjects with high BDI scores excluded from the final sample? Please clarify this point.

Description of experiment 1. Stimuli. “All facial pictures in the system were obtained by all 60 participants who rated their valences and arousal on 9 points”. I’m not sure I understand this sentence. Also, did participants evaluated valence and arousal at the beginning or at the end of the experiment?

Experiment 1. Procedure. If I understand correctly half of the participants were instructed to indicate which one of two faces, presented simultaneously, was the face of a cheater (target: cheater), while the other half had to decide which one was the trustworthy face (target: trustworthy face). But then, the authors state: “The instructions tell the participants, "there are two situations: in one, the one face is a cheater, and the other is trustworthy; in the other, the two faces are trustworthy”. If this is the case that, in some trials, both faces were supposed to be trustworthy, I am wondering what the correct answer for those participants who had to select "the cheater" would have been. Please clarify.

Experiment 1. Results. Following the significant interaction between feedback type and group, I believe post-doc tests between groups could be simple comparisons and not F tests.

Since the task in the encoding/learning phase could have a positive or negative connotation, this being positive when the target is a trustworthy face and negative when the target is a cheater, I am wondering whether one could split the sample based on the type of target and explore whether there are group differences here.

Related to this, I was wondering whether the feedback (exp. 1) might have had a larger effect on the item that was the target of the participant's choice. In other word, if I choose the face to the right as trustworthy (or the cheater) in a given trial, will I remember that face better, as compared to the face on the left? I am not sure if enough trials are available to run such an analysis, but I am curious to know what the authors think about it.

Description of experiment 2. Stimuli. When analyzing valence and arousal scores, now the factor “face type (old, new)” is included. It is not clear to me why this was not the case for experiment 1. Please clarify.

I am somehow missing what exactly were the instructions in the test phase (exp. 1 and 2)? Which key press were available to the subjects? Please specify what would be a “hit” and what a “false alarm”.

It is not clear to me why false alarms are only analyzed in experiment 2.

Please specify if data are made available on a repository.

I think the manuscript will highly benefit from the editing work of a native English.

6. PLOS authors have the option to publish the peer review history of their article (what does this mean?). If published, this will include your full peer review and any attached files.

Reviewer #1: No

Reviewer #2: No

---

## [Author Response · Author response to Decision Letter 0]

14 Mar 2021

Dear Editor and Reviewers:

Thank you for your letter and the reviewers’ comments concerning our manuscript entitled “Effects of two different emotion-inducing methods on the emotional memory of non-clinically depressed individuals” (ID: PONE-D-20-39095). These valuable comments were extremely helpful in revising and improving our manuscript, as well as providing important guidance to our research. We have studied the comments carefully and made corrections that we hope are met with approval. Revisions are marked in blue in the revised edition. The main corrections and responses to reviewers’ comments are as follows:

Responses to comments:

Reviewer #1

Comment 1: What is the relevance of non-clinical depressed subjects for study aimed at understand cognitive/emotional processing in depression? Is there any evidence of a continuum between non-clinical and clinical depression? That high BDI scores in non-clinical populations are a predictor of risk for depression? Since, in several parts of the paper, participants as referred to as “healthy” and “depressed”, it is important point to address this point, at least as a cautionary note in the interpretation of the data.

Response 1: In modern psychiatry, non-clinical depression is a classification that exists between healthy and clinically depressed people. It is also known as subthreshold depression, subclinical depression, or subsyndromal symptomatic depression. In some studies, non-clinical depression was defined as the presence of core symptoms (depressive mood or lack of interest) of depression listed by the DSM-IV, but with fewer than five symptoms or significant functional impairment. Regarding non-clinical depression, between 2 and 5 depressive symptoms and a minimum duration of 2 weeks were required for this diagnosis [1].

Non-clinical depression deserves increased attention, not only because it may be a symptom of a disability that needs treatment but also because patients have a high possibility of experiencing depressive episodes in the future. However, this risk can be avoided through treatment [2]. According to an epidemiological survey, approximately 20~30% of the general population have depressive symptoms. Although most people did not meet the diagnostic criteria for a "depressive episode" set by the DSM-IV, they also have serious damage to their social functions and decline in vocational skills.

In addition, the BDI-II-C has a sensitivity of 91~95%, specificity of 80~85% and positive predictive value of 23~35% for screening adolescent depression and any depressive disorder [3]. The BDI-II-C can easily be adapted to detect major depression in most clinical conditions and recommend an appropriate intervention [4].

Finally, many studies used the BDI to divide non-clinical and healthy participants and used “healthy” and “depressed” to name the two groups [5, 6].

Comment 2: p.9 2nd par (“Stimuli”), line 2: “All facial pictures in the system were obtained by all 60 participants” Unclear, please rephrase.

Response 2: Thank you for your comments. We have revised this sentence.

We revised this part and revised part was marked in blue color in the revised edition. (page 9)

Comment 3: p.9 2nd par (“Stimuli”): Who performed the valence rating of the pictures? The same 60 subjects that participated in the memory experiment or a separate group? If they are the same, when did they perform the rating?

Response 3: Thank you for your comments. The face valence and arousal reported in this study were provided by the Chinese Facial Affective Picture System [7]. The subjects who rated the face pictures were recruited in the study by Wang et al. (2005) [7], but not the subjects in this study. We have revised this section.

We revised this part and revised part was marked in blue color in the revised edition. (page 9)

Comment 4: p.10 1st par, line 2: Judging the “identity” is not exactly what participants were asked to do. It seems more like a personality or trustworthiness judgement was requested.

Response 4: Thank you for your comments. The instructions tell the participants that either one face is a cheater and the other is trustworthy or that the two faces are trustworthy. Participants were asked to judge which one is the cheater. Thus, we considered it to be a task that judges the identity of the face.

Comment 5: p.10 1st par, line 4: “The participants pressed a key to give feedback”. I think ‘to receive feedback’ is meant here, right? Also, as I understand it, on each face pair presentation, there were three possible options to choose from: left is a cheater, right is a cheater, none is a cheater. How did participants express their choice? Vocally? By pressing different keys?

Response 5: We are very sorry for this mistake. The participants guessed the identity of the faces. If participants selected the left face, they pressed the F key. If participants selected the right face, they pressed the J key. The feedback results were given after the key was pressed. We have revised this.

We revised this part and revised part was marked in blue color in the revised edition. (page 10)

Comment 6: p.10 1st par: As I understand it, although I’m not 100 percent sure, the two simultaneously presented training faces were tagged as negative following a “wrong” feedback (or as positive following a “correct” feedback) and both were used in the recognition test. Is that correct?

Response 6: Yes. As the reviewer said, the two simultaneously presented training faces were tagged as negative following a “wrong” feedback (or as positive following a “correct” feedback). All the presented training faces were used in the recognition test. We have added this description.

We revised this part and revised part was marked in blue color in the revised edition. (page 10)

Comment 7: p.10 2nd par, line 2: “The face persisted until the reaction disappeared” Unclear. What do the authors means? That the face remained displayed until the participant responded? Please re-phrase more clearly.

Response 7: Thank you for your comments. We have revised this sentence.

We revised this part and revised part was marked in blue color in the revised edition. (page 10)

Comment 8: p.10 4th par (Results): Several comparisons are made between healthy and “depression”. It should probably read “depressed”

Response 8: Thank you for your comments. We have revised this sentence.

We revised this part and revised part was marked in blue color in the revised edition. (page 11)

Comment 9: p.12 4th par (Stimuli): The authors present the results of an analysis confirming the different valences of the test stimuli, but do not explain how the data were obtained.

Response 9: The faces of the present study come from the Chinese Facial Affective Picture System [7]. In addition to providing faces, the system also provides the data on valences and arousal for each face. The valences and arousal of each face in the system were rated by 60 participants in the study by Wang et al. (2005) [7]. We used the valences and arousal provided by this system for our analysis. We have added this description.

We revised this part and revised part was marked in blue color in the revised edition. (page 13)

Comment 10: p.13 2nd par: Here again, the reader is left to guess what the participants were instructed to do during the learning phase. Please be more explicit.

Response 10: Thank you for your comments. We have added a description.

We revised this part and revised part was marked in blue color in the revised edition. (page 13)

Comment 11: p.14 2nd par, 1st line: “The second experiment was”: I think the authors meant ‘The aim of the second experiment was’.

Response 11: Thank you for your comments. We have revised this sentence.

We revised this part and revised part was marked in blue color in the revised edition. (page 15)

Comment 12: p.15 1st par, line 6-7: I am skeptical about the interpretation of higher RTs in depressed participants: it is not obvious why, when the task is not to identify not the emotion, but to recall the familiarity of a face, a tendency to misinterpret emotions should slow down response times. Furthermore, if a bias to assign a negative emotion to neutral face did contribute to slow down RTs, why are these results opposite to those obtained in Experiment 1 with neutral faces only (i.e. faster RTs in depressed participants)? Actually, the opposite direction of the RT difference between normal and depressed participants between the two experiments is quite puzzling.

Response 12: Thank you for your comments. When emotional and non-emotional information are processed in parallel, information-processing biases could occur as a function of attention to affective information [8, 9]. This is called affective interference. Compared with healthy individuals, depressed individuals react slower to information because they are sensitive to affective interference [8]. Therefore, in the test phase of experiment 2, when participants did non-emotional tasks with the emotional faces, the depressed participants allocated more cognitive resources to emotion. In turn, this caused the dispersion of cognitive resources from non-emotional tasks [10, 11, 12]. Thus, the reaction time of depressed participants was longer than that of healthy participants. We have revised this part.

We revised this part and revised part was marked in blue color in the revised edition. (page 16)

Comment 13: p.15 2nd par, line 3-4: “on the performance of memory”: ‘on memory performance’ wound read better.

Response 13: Thank you for your comments. We have revised this sentence.

We revised this part and revised part was marked in blue color in the revised edition. (page 16)

Comment 14: p.16 1st par line 7: “individuals with depression have a stronger attention bias toward negative materials and the further promotion of subsequent memory effects” The second part of this sentence is unclear. Please rephrase.

Response 14: Thank you for your comments. We have revised this sentence.

We revised this part and revised part was marked in blue color in the revised edition. (page 17)

Comment 15: Finally, the manuscript must be checked by an English language native speaker.

Response 15: Thank you for your comments. The language of our manuscript have been refined and polished by a professional editing company.

1. Rodríguez MR, Nuevo R, Chatterji S, Ayuso-Mateos JL. (2012). Definitions and factors associated with subthreshold depressive conditions: a systematic review. Bmc Psychiatry. 2012; 12: 181.

2. Cuijpers P, Beekman ATF, Reynolds CF. (2012). Preventing depression: a global priority. JAMA the Journal of the American Medical Association. 2012; 307(10): 1033-4.

3. Yang WH, Xiong G. Screening for Adolescent Depression: Validity and Cut-off Scores for Depression Scales. Chinese Journal of Clinical Psychology. 2016; 24(6):1010-1015.

4. Wang YP, Gorenstein C. Assessment of depression in medical patients: a systematic review of the utility of the beck depression inventory-ii. Clinics. 2013; 68(9): 1274-1287.

5. Li P, Song XX, Wang J, Zhou XR, Li JY, Lin FT et al. Reduced sensitivity to neutral feedback versus negative feedback in subjects with mild depression: Evidence from event-related potentials study. Brain & Cognition. 2015; 100: 15-20.

6. Goddard L, Dritschel B, Burton A. Social problem solving and autobiographical memory in non-clinical depression. The British Journal of Clinical Psychology. 1997; 36(3): 449-451.

7. Wang Y, Luo YJ. Standardization and Assessment of College Students' Facial Expression of Emotion. Chinese Journal of Clinical Psychology. 2005; 13(4): 396-398.

8. Siegle GJ, Ingram RE, Matt GE. Affective Interference: An Explanation for Negative Attention Biases in Dysphoria. Cognitive Therapy and Research. 2002; 26(1): 73-87.

9. Siegle GJ. A neural network model of attention biases in depression. Progress In Brain Research. 1999; 121(5): 407-432.

10. Bistricky SL, Ingram RE, Atchley RA. Facial affect processing and depression susceptibility: cognitive biases and cognitive neuroscience. Psychological Bulletin. 2011; 137(6):998-1028.

11. Kellough JL, Beevers CG, Ellis, AJ, Wells TT. Time course of selective attention in clinically depressed young adults: An eye tracking study. Behaviour Research and Therapy. 2008; 46(11): 1238-1243.

12. Peckham AD, McHugh RK, Otto MW. A meta-analysis of the magnitude of biased attention in depression. Depression and Anxiety. 2010; 27(12): 1135-1142.

Reviewer #2 

Comment 1: Subjects were divided in two groups (healthy participants vs. non-clinical depression participants) based on BDI scores: “participants with BDI-II-C scores greater than or equal to 14 were selected as the depression group (non-clinical).” How do the authors know that the group of “depressed” participants did not include subjects with clinical/severe depression? Did the authors collected more information about the subjects’ medical history and were subjects with high BDI scores excluded from the final sample? Please clarify this point.

Response 1: Thank you for your comments. The participants were not diagnosed with clinical depression and did not use anti-depressants. They also did not have comorbidities or use other psychotropic drugs. This information was specified within the recruitment advertisement as recruitment criteria. There were no participants with BDI-II-C scores greater than or equal to 29. We have added the descriptions to the manuscript.

We revised this part and revised part was marked in blue color in the revised edition. (page 9)

Comment 2: Description of experiment 1. Stimuli. “All facial pictures in the system were obtained by all 60 participants who rated their valences and arousal on 9 points”. I’m not sure I understand this sentence. Also, did participants evaluated valence and arousal at the beginning or at the end of the experiment?

Response 2: Thank you for your comments. The faces in the present study come from the Chinese Facial Affective Picture System [1]. In addition to providing faces, the system also provides the data for valences and arousal for each face. The valences and arousal of each face in the system were rated by 60 participants in the study of Wang et al. (2005) [1]. We used the valences and arousal data provided by this system for our analysis. Participants of the present study did not evaluate valence or arousal at the beginning or end of the experiment because it may have affected their memory performance or results of the evaluation. We have added a description and revised this sentence.

We revised this part and revised part was marked in blue color in the revised edition. (page 9)

Comment 3: Experiment 1. Procedure. If I understand correctly half of the participants were instructed to indicate which one of two faces, presented simultaneously, was the face of a cheater (target: cheater), while the other half had to decide which one was the trustworthy face (target: trustworthy face). But then, the authors state: “The instructions tell the participants, "there are two situations: in one, the one face is a cheater, and the other is trustworthy; in the other, the two faces are trustworthy”. If this is the case that, in some trials, both faces were supposed to be trustworthy, I am wondering what the correct answer for those participants who had to select "the cheater" would have been. Please clarify.

Response 3: Thank you for your comments. In either case, participants must choose a face that is more likely to be trustworthy (or more likely to be a cheater). If this is the case that, in some trials, both faces were supposed to be trustworthy. Participants must make a force-choice. We have added a description.

We revised this part and revised part was marked in blue color in the revised edition. (page 10)

Comment 4: Experiment 1. Results. Following the significant interaction between feedback type and group, I believe post-doc tests between groups could be simple comparisons and not F tests.

Response 4: Thank you for your comments. We have revised this.

We revised this part and revised part was marked in blue color in the revised edition. (page 11)

Comment 5: Since the task in the encoding/learning phase could have a positive or negative connotation, this being positive when the target is a trustworthy face and negative when the target is a cheater, I am wondering whether one could split the sample based on the type of target and explore whether there are group differences here.

Response 5: Thank you for your comments. We have split the data and analyzed it. However, there was no significant difference between the two groups and interactions. Because this was not included in the aim of the present study, we did not present the results of that analysis.

Comment 6: Related to this, I was wondering whether the feedback (exp. 1) might have had a larger effect on the item that was the target of the participant's choice. In other word, if I choose the face to the right as trustworthy (or the cheater) in a given trial, will I remember that face better, as compared to the face on the left? I am not sure if enough trials are available to run such an analysis, but I am curious to know what the authors think about it.

Response 6: Thank you for your comments. As the reviewer said, we also believe that the chosen faces would associate with better memory. However, both faces were used in the recognition test. Thus, the effect was avoided and this would not affect results of the study. Also, due to the experimental programming, we were unable to analyze the differences.

Comment 7: Description of experiment 2. Stimuli. When analyzing valence and arousal scores, now the factor “face type (old, new)” is included. It is not clear to me why this was not the case for experiment 1. Please clarify.

Response 7: All faces were neutral in experiment 1. Emotional conditions were manipulated by feedback results. New faces were neutral and needn’t be divided into three conditions in the test phase. Therefore, the faces were divided into four conditions (positive, negative, neutral, new). We conducted one-way ANOVA (condition: positive, negative, neutral, new) for the valences and arousal. We wanted to ensure there was no difference in valences and arousal between the four conditions. Yet, in experiment 2, the emotional conditions were divided by expression. If all of the new faces were neutral, it would be too easy for participants to judge faces as the old ones would include all of the positive or negative faces. Therefore, new faces also had three emotional conditions in the test phase. We then performed a two-way ANOVA 3 (valence: positive, negative, or neutral) × 2 (face type: new and old) for the valences and arousal.

Comment 8: I am somehow missing what exactly were the instructions in the test phase (exp. 1 and 2)? Which key press was available to the subjects? 

Response 8: Thank you for your comments. If participants judged that it was new, they pressed the F key. If participants judged that it was old, they pressed the J key. The button response was counterbalanced across participants. We have added a description.

We revised this part and revised part was marked in blue color in the revised edition. (page 10; page 14)

Comment 9: Please specify what would be a “hit” and what a “false alarm”. It is not clear to me why false alarms are only analyzed in experiment 2.

Response 9: Thank you for your comments. We have added the description of “hit” and “false alarm” to the manuscript. All faces in experiment 1 were neutral. Different conditions were divided by feedback results. New faces in the test phase were also neutral, and thus, did not need to be divided into three conditions. Therefore, there was no difference in the false alarm rate among the conditions. In contrast to experiment 1, different conditions were divided by expression in experiment 2. If all of the new faces were neutral, it would be too easy for participants to judge faces as the old ones would include all of the positives or negatives. Therefore, new faces of the three conditions represented three kinds of expression, and the false alarm rates in the three conditions may have been different. Thus, in experiment 2, the Pr value, reaction time, hit rate, and false alarm rate were analyzed. We revised the description of the analysis.

We revised this part and revised part was marked in blue color in the revised edition. (page 11; page 14)

Comment 10: Please specify if data are made available on a repository.

Response 10: Thank you for your comments. We have addressed it.

Comment 11: I think the manuscript will highly benefit from the editing work of a native English.

Response 11: Thank you for your comments. The language of our manuscript have been refined and polished by a professional editing company.

1. Wang Y, Luo YJ. Standardization and Assessment of College Students' Facial Expression of Emotion. Chinese Journal of Clinical Psychology. 2005; 13(4): 396-398.

We have carefully studied the reviewers’ suggestions and made corresponding amendments in order to improve the manuscript. All changes are being revised and marked in blue. 

We earnestly appreciate the Editor and Reviewers’ helpful work and hope that the revised manuscript will be met with approval.

Once again, thank you very much for your comments and suggestions!

---

## [Decision Letter · Decision Letter 1]

26 Mar 2021

Effects of two different emotion-inducing methods on the emotional memory of non-clinically depressed individuals

PONE-D-20-39095R1

Dear Dr. Mo,

We’re pleased to inform you that your manuscript has been judged scientifically suitable for publication and will be formally accepted for publication once it meets all outstanding technical requirements.

Kind regards,

Angela Sirigu

Academic Editor

PLOS ONE

Reviewers' comments:

Reviewer's Responses to Questions

**Comments to the Author**

1. If the authors have adequately addressed your comments raised in a previous round of review and you feel that this manuscript is now acceptable for publication, you may indicate that here to bypass the “Comments to the Author” section, enter your conflict of interest statement in the “Confidential to Editor” section, and submit your "Accept" recommendation.

Reviewer #1: All comments have been addressed

Reviewer #2: All comments have been addressed

2. Is the manuscript technically sound, and do the data support the conclusions?

Reviewer #1: Yes

Reviewer #2: Yes

3. Has the statistical analysis been performed appropriately and rigorously? 

Reviewer #1: Yes

Reviewer #2: Yes

4. Have the authors made all data underlying the findings in their manuscript fully available?

Reviewer #1: Yes

Reviewer #2: Yes

5. Is the manuscript presented in an intelligible fashion and written in standard English?

Reviewer #1: Yes

Reviewer #2: Yes

6. Review Comments to the Author

Reviewer #1: The authors responded satisfactorily to both reviewers' remarks and made the requested changes in the revised manuscript. I have no further comments

Reviewer #2: The authors have successfully addressed the comments I made to the previous version of the manuscript.

7. PLOS authors have the option to publish the peer review history of their article (what does this mean?). If published, this will include your full peer review and any attached files.

Reviewer #1: No

Reviewer #2: No

---

## [Editor Report · Acceptance letter]

17 May 2021

PONE-D-20-39095R1 

Effects of two different emotion-inducing methods on the emotional memory of non-clinically depressed individuals 

Dear Dr. Mo:

I'm pleased to inform you that your manuscript has been deemed suitable for publication in PLOS ONE. Congratulations! Your manuscript is now with our production department. 

Kind regards, 

on behalf of

Dr. Angela Sirigu 

Academic Editor

PLOS ONE